# Numerical Analysis of Stable (FAPbI_3_)_0.85_(MAPbBr_3_)_0.15_-Based Perovskite Solar Cell with TiO_2_/ZnO Double Electron Layer

**DOI:** 10.3390/nano13081313

**Published:** 2023-04-08

**Authors:** Yongjin Gan, Guixin Qiu, Binyi Qin, Xueguang Bi, Yucheng Liu, Guochao Nie, Weilian Ning, Ruizhao Yang

**Affiliations:** 1Guangxi Colleges and Universities Key Laboratory of Complex System Optimization and Big Data Processing, Yulin Normal University, Yulin 537000, China; 2Office of the Party Committee, Guangxi Minzu Normal University, Chongzuo 532200, China; 3School of Physics and Telecommunication Engineering, Yulin Normal University, Yulin 537000, China; 4Optoelectronic Information Research Center, School of Physics and Telecommunication Engineering, Yulin Normal University, Yulin 537000, China; 5Department of Mechanical Engineering, South Dakota State University, Brookings, SD 57007, USA

**Keywords:** perovskite solar cell, energy level, SCAPS-1D, defect density, temperature

## Abstract

Although perovskite solar cells have achieved excellent photoelectric conversion efficiencies, there are still some shortcomings, such as defects inside and at the interface as well as energy level dislocation, which may lead to non-radiative recombination and reduce stability. Therefore, in this study, a double electron transport layer (ETL) structure of FTO/TiO_2_/ZnO/(FAPbI_3_)_0.85_(MAPbBr_3_)_0.15_/Spiro-OMeTAD is investigated and compared with single ETL structures of FTO/TiO_2_/(FAPbI_3_)_0.85_(MAPbBr_3_)_0.15_/Spiro-OMeTAD and FTO/ZnO/(FAPbI_3_)_0.85_(MAPbBr_3_)_0.15_/Spiro-OMeTAD using the SCAPS-1D simulation software, with special attention paid to the defect density in the perovskite active layer, defect density at the interface between the ETL and the perovskite active layer, and temperature. Simulation results reveal that the proposed double ETL structure could effectively reduce the energy level dislocation and inhibit the non-radiative recombination. The increases in the defect density in the perovskite active layer, the defect density at the interface between the ETL and the perovskite active layer, and the temperature all facilitate carrier recombination. Compared with the single ETL structure, the double ETL structure has a higher tolerance for defect density and temperature. The simulation outcomes also confirm the possibility of preparing a stable perovskite solar cell.

## 1. Introduction

Energy depletion is one of the major challenges of our time. Many researchers worldwide have been working hard to address the urgent demand for novel sustainable energy sources in the last decades. Among a variety of sustainable energy options, solar energy has emerged as the most promising choice for the plan of a concretely sustainable world [1]. Solar cells, which convert ecologically friendly and inexhaustible solar energy into electrical power, are expected to meet all the global energy demand [2]. Perovskite solar cells (PSCs) are one of the most promising technologies for next generation photovoltaics [3] due to their many superior optical and electronic properties, such as their low costs, simple manufacturing, excellent optical absorption, high carrier mobilities, and large diffusion lengths [4,5]. Moreover, the solution-based preparation of perovskite thin films is rather cost-effective as it only requires low temperatures and small amounts of materials [6]. Since the organic-inorganic hybrid PSCs were first proposed in 2009 [7], the power conversion efficiency (*PCE*) values of single-junction PSCs have increased from 3.8% to 25.7% [8,9,10,11,12], which is comparable to that of C-Si solar cells. However, there is still a certain gap between the current *PCE* level and the theoretical limit of *PCE* of PSCs [2] due to the nonradiative recombination of carriers caused by the point defects of the materials, light emitting, or the heating effect [13]. To prepare high-quality films, and enhance the *PCE* of PSCs, various methods have been proposed in previous studies, such as passivation defects [14], interface modification [15], optimize transmission materials [16] and component substitution [17]. However, the promotion and application of PSCs have been hindered because of some disadvantages, including toxicity, degradation mechanism, poor long-time stability [18], and hysteresis [19,20]. PSCs are extremely sensitive to moisture, oxygen, temperature, and ultraviolet (UV) light [18], which would cause their degradation. The degradation of PSCs mainly involves perovskite material degradation and interface degradation. The photo perovskite material degradation could result in thermal instability, ion movement, and moisture instability [21]. *PCE* is inhibited by UV light due to deep electronic trap energies at the oxygen vacancies in TiO_2_ [22], a common electron transport layer (ETL). Interestingly, interface stability of PSCs can be enhanced by controlling the defect density of the perovskite and interfacial layers. The methods commonly used to control the defect density include adopting novel deposition and coating techniques, novel carrier transport layers, compositional engineering, and encapsulation methods [23]. Research shows that stability and *PCE* can be improved at the same time by mixing cation perovskites [24] such as mixing FAPbI_3_ and MAPbBr_3_ perovskites with different ion diffusion lengths [25]. Because MA has a dipole moment 10 times larger than that of FA, the incorporation of a small amount of MAPbBr_3_ would significantly improve the stability of the structure [26]. Reyna et al. [27] fabricated devices by mixing perovskites containing FA and MA cations, and experimental results confirmed the excellent stability of mixed MA/FA mesoscopic PSCs. In 2017, Li et al. [28] prepared high-quality (FAPbI_3_)_0.85_(MAPbBr_3_)_0.15_ perovskite films by a solvent engineering method, and *PCE* increased from 15.3% to 16.8%. Paek et al. [29] prepared efficient and stable PSCs combined with the mixed perovskite (FAPbI_3_)_0.85_(MAPbBr_3_)_0.15_ and they obtained an excellent *PCE* of 18.9%. Bencherif et al. [30] optimized an (FAPbI_3_)_1−x_(MAPbBr_3_)_x_-based PSC using a numerical method and confirmed that a high *PCE* could be obtained theoretically. Recently, Miao et al. [31] proposed a strategy to achieve efficient and stable perovskite solar cells via in situ growth of ultra-thin perovskitoid layers. Gong et al. [32] suggested a buried interface stabilization strategy based on synergistic effect of fluorine and sulfonyl functional groups to raise the stability and efficiency of PSCs. Wang et al. [33] employed nickel iodide to passivate iodine vacancies and suppress non-radiative recombination, thereby obtaining inorganic perovskite/silicon tandem solar cells with an enhanced efficiency of 23%.

Due to the limitations of the preparation process and the type of carrier transport layer materials, there are interface defects and energy level dislocations in current PSCs, which lead to non-radiation recombination and reduce the open-circuit voltage (*V_oc_*) and filling factor (*FF*) [15,34]. The presence of these defects at the interface between the ETL and perovskite active layer is an inducement for the non-radiation recombination. The conduction band offset (*CBO*) is defined as the difference between the conduction band level of the ETL and that of the perovskite, and carrier recombination at the interface is affected by the *CBO*. Double ETL structures can effectively inhibit the carrier recombination. A double ETL structure is formed by introducing another ETL material between the ETL and the perovskite active layer. PSCs with double ETLs can provide excellent performances by reducing the non-radiation recombination [35,36]. Outcomes of relevant research have also confirmed this phenomenon. For example, Qiu et al. [37] reported a high-efficiency PSC with a PCBM/ZnO double ETL. The researchers found that the additional ZnO not only improves the energy level alignment, but also enhances the air stability of the device. Wu et al. [38] confirmed that using a PSC with TiO_2_/SnO_x_Cl_y_ as its double ETL structure would drastically improve *PCE* and reduce hysteresis. Wang et al. [39] adopted a ZnO/SnO_2_ double ETL structure to improve the *PCE* value to 19.1%. Ren et al. [40] reported a high-performance planar heterojunction PSC with PCBM/N2200 as a double ETL structure. These investigators believed that the built-in potential of the device was enhanced and the interfacial energy barrier of MAPbI_3_/PCBM was reduced, leading to a higher *V_oc_*. Wang et al. [36] demonstrated that the *PCE* value of the SnO_2_/TiO_2_ double ETL structure was improved by 2% because the carrier transfer process was improved and carrier recombination at the interface was suppressed. The double ETL structure could improve the conduction band energy level matching between the transport layer and the perovskite active layer, resulting in a reduction of carrier recombination at the interface between the perovskite active layer and the transport layer.

This work aimed to suggest possible optimization strategies to improve the *PCE* and stability of (FAPbI_3_)_0.85_(MAPbBr_3_)_0.15_-based PSCs using the solar cell capacitance simulator (SCAPS-1D). In this study, the influence of *CBO* on the performance of a single ETL structure was first analyzed, and the effect of the difference between the ETL conduction band level and the cathode work function on the cell performance was examined. Then, the effect of the defect densities in the perovskite active layer and at the interface between the ETL and the perovskite active layer was studied for both single ETL and double ETL structures. Finally, the effect of the working temperature on the performances of the single ETL structure was compared with that of the double ETL structure. The mechanism analysis of high-performance PSCs based on the double ETL structure has great significance for the construction of high-efficiency and stable photovoltaic devices.

## 2. Simulation and Modelling

### 2.1. Numerical Method

Numerical simulation methods are helpful to understand the deep physical mechanism of PSCs, provide guidance for experimental preparation, and accelerate the research of PSCs. A one-dimensional software SCAPS-1D was adopted as the numerical simulation platform in this study. SCAPS-1D was developed by Professor Burgelman of Ghent University in Belgium [41] and is being widely used for numerical modeling and simulation based on fundamental equations of semiconductor device physics. Under certain boundary conditions of the semiconductor device, SCAPS-1D can be used to determine *J-V* characteristics, spectral response, capacitance-frequency characteristics, capacitance–voltage characteristics by solving Poisson equation and electron and hole continuity equations. Poisson equation, electron and hole continuity equations, and drift-diffusion equations are listed as follows [42]:(1)$$\frac{\partial^{2}\varphi }{\partial^{2}x^{2}}=\frac{q}{\varepsilon }\left(n-p\right)$$
(2)$$\frac{\partial n}{\partial t}=\frac{1}{q}\frac{\partial J_{n}}{\partial x}+G-R,\;\frac{\partial n}{\partial t}=-\frac{1}{q}\frac{\partial J_{p}}{\partial x}+G-R$$
(3)$$J_{n}=qD_{n}\frac{\partial n}{\partial x}-q\mu_{n}n\frac{\partial \varphi }{\partial x},\;J_{p}=qD_{p}\frac{\partial p}{\partial x}-q\mu_{p}p\frac{\partial \varphi }{\partial x}\;$$

In the equations above, *ε* is the dielectric constant, *q* is the electronic charge, *G* is the generation rate, *R* is the recombination rate, *D* is the diffusion coefficient, *φ* is electrostatic potential, and *p* and *n* are free holes and electrons concentration, respectively.

Under normal circumstances, these nonlinear differential equations (Equations (1)–(3)) are difficult to solve. However, if the boundary conditions are specified, device performances such as *J-V* characteristics, spectral response, capacitance–frequency characteristics and capacitance–voltage characteristics can be obtained. The potential at the back contact is set to 0 V, and the boundary conditions at the front contact are defined as follows:(4)$$\emptyset \left(0\right)=\emptyset_{f}-\emptyset_{b}+V_{app}$$
(5)$$J_{n}\left(0\right)=qS_{n}^{f}\left[n^{\prime }\left(0\right)-n_{eq}\left(0\right)\right]$$
(6)$$J_{p}\left(0\right)=-qS_{n}^{f}\left[p^{\prime }\left(0\right)-p_{eq}\left(0\right)\right]$$

The boundary conditions at the back contact are:(7)$$\emptyset \left(L\right)=0$$
(8)$$J_{n}\left(L\right)=qS_{n}^{b}\left[n^{\prime }\left(L\right)-n_{eq}\left(L\right)\right]$$
(9)$$J_{p}\left(L\right)=qS_{p}^{b}\left[p^{\prime }\left(L\right)-p_{eq}\left(L\right)\right]$$

In the above equations, the labels f and b represent the front and the back electrode, respectively. *φ* represents the work function of electrode, and *V*_app_ denotes the bias voltage applied to the device or the bias voltage generated by the incident light. *p*_eq_ and *n*_eq_, respectively, denote the hole and electron concentrations at the electrode interface under thermal equilibrium conditions. *p*′ and *n*′ are carrier concentrations at the front and back electrodes, respectively. $S_{f_{p}}$, $S_{f_{n}^{\prime }}$, $S_{p}^{b}$, and $S_{n}^{b}$, represent the surface recombination rates of electrons and holes at the electrodes, respectively, which are related to the surface passivation.

### 2.2. Structure and Band Diagram

To compare a traditional single ETL device with a double ETL device, three PSCs, FTO/TiO_2_/(FAPbI_3_)_0.85_(MAPbBr_3_)_0.15_/Spiro-OMeTAD, FTO/ZnO/(FAPbI_3_)_0.85_(MAPbBr_3_)_0.15_/Spiro-OMeTAD and FTO/TiO_2_/ZnO/(FAPbI_3_)_0.85_(MAPbBr_3_)_0.15_/Spiro-OMeTAD, were constructed in this study, as shown in Figure 1. For the convenience of subsequent discussion, these three devices are referred to as device 1, device 2, and device 3. It can be seen from Figure 1 that (FAPbI_3_)_0.85_(MAPbBr_3_)_0.15_ is a photoactive layer, and Spiro-OMeTAD is a hole transport layer (HTL). FTO and Au are used as the front contact electrode and the rear contact electrode, respectively. For device 1 and device 2, TiO_2_ and ZnO are adopted as the ETL, respectively, while the ETL of device 3 is a double-layer structure composed of TiO_2_ and ZnO. The total thickness of the ETL remains the same in the three devices. As discovered by Benami et al. [43], the use of ZnO as an electron transport material could improve the cell efficiency of PSCs compared to TiO_2_.

Energy level diagrams of the materials involved are shown in Figure 2. As shown in that figure, (FAPbI_3_)_0.85_(MAPbBr_3_)_0.15_ absorbs photons with energies greater than its band gap and generates excitons. Then, the excitons are separated at the interface between the perovskite active layer and the carrier transport layers, forming free electrons and free holes. Free electrons and free holes continue to move along the ETL and HTL, respectively. It can be seen from Figure 2 that the conduction band of (FAPbI_3_)_0.85_(MAPbBr_3_)_0.15_ is higher than those of TiO_2_ and ZnO, which facilitates the transmission of electrons. However, the valence bands of TiO_2_ and ZnO are lower than that of (FAPbI_3_)_0.85_(MAPbBr_3_)_0.15_, which is conducive to preventing the holes from moving along the ETL direction and avoiding carrier recombination. Similarly, the conduction band and valence band of Spiro-OMeTAD are higher than those of (FAPbI_3_)_0.85_(MAPbBr_3_)_0.15_, which is conducive to the movement of holes and hindered the transmission of electrons. A stepped conduction band structure was formed in device 3, resulting in a decrease in the non-radiative recombination caused by the energy level dislocation. Therefore, from the analysis of the energy band structure, the cell devices constructed in this paper are feasible.

### 2.3. Basic Parameters of Devices

The basic physical parameters of different materials listed in Table 1 were selected from relevant previous studies [27,44,45,46,47,48,49,50]. To make the results easy to compare, the thickness of ETLs remains the same (100 nm); the thickness of TiO_2_ in device 1 and the thickness of ZnO in device 2 are both 100 nm; and the thickness of TiO_2_ and ZnO in device 3 are both 50 nm. In addition, the electron and hole thermal velocity were set to 10^7^ cm/s and the value of the absorption coefficient *α* was calculated by setting the pre-factor *A*_α_ to be 10^5^ cm^−1^ eV^−0.5^ using the following equation:(10)$$\alpha =A_{\alpha }{\left(hv-E_{g}\right)}^{1/2}$$

The parameters of the interface between the perovskite active layer and the carrier transport layers are shown in Table 2. The simulation process was carried out under AM1.5G solar lighting and a 100 mW/cm^2^ incident power density.

In MAPbBr_3_, the diffusion lengths of electrons (*L*_e-MAPbBr3_) and holes (*L*_h-MAPbBr3_) are 0.83 µm and 53.5 µm [51], respectively. The diffusion lengths of electrons (*L*_e-FAPbI3_) and holes (*L*_h-FAPbI3_) are 0.177 µm and 0.813 µm in FAPbI_3_, respectively [52]. The diffusion lengths of electrons (*L*_e_) and holes (*L*_h_) in (FAPbI_3_)_0.85_(MAPbBr_3_)_0.15_ are calculated as follows [49]:(11)$$L_{e}=0.85L_{e-FAPbI_{3}}+0.15L_{e-FAPbBr_{3}}$$
(12)$$L_{h}=0.85L_{h-FAPbI_{3}}+0.15L_{h-FAPbBr_{3}}$$

Therefore, the values of *L*_e_ and *L*_h_ are calculated to be 0.275 µm and 8.716 µm, respectively. The value of *L*_h_ is in good agreement with the experimental results [53]. The carriers’ mobility is related to the trap density and diffusion length through the following equations:(13)$$\tau =1/\left(\sigma V_{th}N_{t}\right)$$
(14)$$L=\sqrt{\left(\frac{K_{B}T}{q}\right)\mu \tau }$$

Thus, the carrier mobility *μ* of the mixed cation PSCs can be obtained by introducing parameters including Boltzmann constant *K*_B_, working temperature *T*, carrier capture cross section *σ*, and the thermal velocity of electrons and holes *V*_th_.

The performances of the three initial devices constructed using the abovementioned parameters are shown in Figure 3. It is obviously observed that the *J-V* characteristic curve and *PCE* of device 1 are relatively poor, and *J_sc_* of device 1 is slightly lower than the other two devices. Quantum efficiency (*QE*) measures the absorption of photons by devices. It can be seen from Figure 3c that the *QE* of device 1 is slightly lower than the other two devices, indicating that the absorption of photons by device 1 is inadequate; therefore, *J_sc_* of device 1 is relatively low. Moreover, Figure 3d’s conduction band of device illustrates that the interface energy band shift between the ETL and perovskite active layer of device 1 is the largest (the area circled by the green dotted line), which affects the transportation and extraction of electrons, then affects carrier recombination, *V_oc_* and *PCE*. Similarly, Figure 3d shows that a Schottky barrier is formed between the ETL and the cathode in device 2 (the area circled by the orange dotted line); the successful collection of electrons requires more energy compared with device 3.

From those figures, it can be deduced that the ETL is a major contributor to the interface modification, controlling the carrier recombination rate and also aligning the inter-layer energy levels. A proper ETL is helpful to conduct electrons and block holes and reduce energy loss. Thus, it should have a high conductivity and an excellent band alignment with the perovskite active layer [54]. Device with double ETL can improve the energy level alignment and promote electron extraction, thus achieving high performance.

## 3. Results and Discussion

### 3.1. Analysis of Device with TiO_2_ Single Electron Transport Layer

This section discusses the performance of the device with the TiO_2_ single electron transport layer. First, the effect of the electron affinity of TiO_2_ on the cell performance is discussed. The energy band structure of TiO_2_ could be changed by a dopant. Therefore, the electron affinity of TiO_2_ was set in the range of 4.1–4.5 eV in this study, and the simulation results are shown in Figure 4. It can be seen from Figure 4a that, with a gradual increase in the electron affinity of TiO_2_, *V_oc_* shows a gradual downward trend, but the short-circuit current density (*J_sc_*) changes slightly. Figure 4b shows that the *PCE* decreases with the increase in the electron affinity of TiO_2_.

*CBO* is defined as the difference between the conduction band level of the ETL and that of the perovskite, and the schematic diagram of *CBO* is shown in Figure 5. The value of *CBO* equals to the difference between the conduction band level of the ETL (*E_c_ETL_*) and he conduction band level of perovskite (*E_c_PVSK_*), *CBO* = *E_c_ETL_* − *E_c_PVSK_*. It can be seen from Table 2 that the value of *E_c_PVSK_* is 4.05 eV, while the electron affinity of TiO_2_ varies from 4.1 eV to 4.5 eV. Therefore, the *CBO* values are negative. If the *CBO* values are negative, an energy cliff could be formed at the ETL/perovskite interface, as shown in Figure 6a. The energy cliff does not hamper the photo-generated electron flow. Therefore, it would not have much impact on *J_SC_*. Thus, *J_SC_* fluctuates slightly with the change in the electron affinity of TiO_2_. With the increase in the electron affinity of TiO_2_, carrier recombination is promoted, and the simulation results are shown in Figure 6b. The increasing carrier recombination rate leads to the continuous reduction of *V_OC_*. Therefore, *V_OC_* would decrease with the increase in the electron affinity of TiO_2_. The decrease in *V_OC_* also would lead to a decrease in *PCE*. In another word, the performance decreases with the increase in the electron affinity of TiO_2_. Therefore, reducing the conduction band offset between the perovskite and ETL would improve the cell performance [55].

Next, the influence of the relationship between the conduction band level of the ETL and the cathode work function on the cell performance is analyzed. *Φ* is defined as the difference between the conduction band level of the ETL and the cathode work function, and a schematic diagram of *Φ* is shown in Figure 5. The value of *Φ* is changed when the electron affinity of TiO_2_ changes. The *J-V* characteristic curves and *PCE* with *Φ* varying from 0.2 eV to 0.6 eV are shown in the Figure 7. It can be clearly seen that with the increase in *Φ*, the cell performance decreases monotonically.

Figure 8a illustrates that the energy band slope of the perovskite active layer varies as *Φ* decreases from 0.6 eV to 0.2 eV, indicating that the built-in voltage *V_bi_* continuously decreases. *V_bi_* is closely related to the carrier extraction. The larger the value of *V_bi_*, the more efficient the carrier extraction. Therefore, it can be inferred from Figure 8a that the carrier extraction is gradually inhibited with an increase in *Φ*. Due to the reduction of *V_bi_*, the efficiency of photogenerated carrier extraction decreases and the concentration of electrons in the perovskite active layer increases, as shown in Figure 8b, leading to an increase in the internal carrier recombination rate of the perovskite active layer. Eventually, the cell performance will gradually deteriorate.

From the abovementioned analysis, with the continuous increases in *CBO* and *Φ*, the performance of the cell has a downward trend. To obtain a better cell performance, *CBO* and *Φ* should not be too large. However, it can be seen from Figure 5 that when *CBO* is reduced, *Φ* increases and vice versa. It is difficult to control the magnitudes of *CBO* and *Φ* simultaneously within the traditional structure. However, the double ETL structure could reduce the conduction band offset between the perovskite active layer and the ETL. The structure contains two different ETLs (ETL_1 and ETL_2), which are shown in Figure 9. The *CBO* is reduced by increasing the electron affinity of ETL_1; and *Φ* is reduced by increasing the electron affinity of ETL_2. Therefore, by adjusting the electron affinity of ETL_1 and ETL_2 separately, *CBO* and *Φ* can be reduced simultaneously. Reducing the conduction band offset could reduce the non-radiative recombination at the interface and meanwhile reduce the difference between the ETL and the cathode work function, promoting the carrier transport, thus improving the cell performance [55].

### 3.2. Effects of Defect Density of Perovskite Active Layer on Performance

As an optical absorption layer, the generation and recombination of carriers mainly occur in the perovskite active layer. Therefore, it is of great significance to study the effect of the defect density of the perovskite active layer *N_t_* on the device performance for improving *PCE*. The morphology and quality of the perovskite active layer film directly affect the performances of PSCs. If the film has poor quality, a high defect density will be formed in the perovskite active layer and the carrier recombination rate will increase, resulting in a decline in the performance. To explore the effect of *N_t_* on the cell performance, *N_t_* was set to change within the range of 10^13^–10^17^ cm^−3^. The variation trend of the *J-V* characteristic curve is shown in Figure 10, and the variations of the cell output parameters with *N_t_* are shown in Figure 11. Based on the results in Figure 11, the decrease ratio of the output parameters is listed in Table 3. As shown in Figure 10, with the increase in *N_t_*, the *J-V* characteristic curve deteriorates. Similarly, Figure 11 shows that the cell output parameters, including *V_oc_*, *J_sc_*, *FF* and *PCE,* also decrease with the increase in *N_t_*. Table 3 shows that the output parameters of device 3 fluctuate the least with the change in *N_t_*. The simulation results show that the performance of device 3 is at a higher level for the different *N_t_* values compared with the performances of device 1 and device 2. With the change in *N_t_*, the performance fluctuation of device 3 is weaker than those of device 1 and device 2.

Relevant research [56] showed that defects in the perovskite active layer led to carrier recombination through trap states. This type of recombination is called as trap-assisted Shockley–Read–Hall (SRH) recombination and can be expressed as follows [57,58,59]:(15)$$R^{SRH}=\frac{np-n_{i}^{2}}{\tau_{p}\left(n+n_{i}\right)+\tau_{n}\left(p+p_{i}\right)}$$

The relationship between the carrier diffusion length *L* and the carrier lifetime *τ* can be described by the following equation:(16)$$L=\sqrt{D\tau }$$
where *D* is the diffusion coefficient and is given by:(17)$$D=\frac{K_{B}T}{q}\mu$$

Finally, *L* can be expressed by the following equation:(18)$$L=\sqrt{\frac{K_{B}T\mu }{q\sigma V_{th}N_{t}}}$$

It can be seen from Equation (18) that as *N_t_* increases, *L* decreases. The decrease in *L* makes *τ* smaller, resulting in an enhancement of *R^SRH^*. Therefore, the increase in *N_t_* leads to the formation of more carrier recombination centers, which promotes carrier recombination (Figure 12); therefore, the recombination current in the perovskite active layer increases (Figure 13), which inhibits *V_oc_*. In another word, the cell performance deteriorates with the increase in the defect density of the perovskite active layer.

Table 3 clearly shows that the decreases in the output parameters of device 3 are the smallest among the three devices. It can be seen from Figure 11 that, when other conditions remain the same and only the size of *N_t_* is changing, *R^SRH^* of the three devices would increase with the increase in *N_t_*. In comparison, the *R^SRH^* of device 1 changes greatly, while the *R^SRH^* of device 3 changes only slightly and remains at a relatively low level. Similarly, the same trend is presented in Figure 14, in which the red and the blue lines represent the *J-V* characteristic curves when *N_t_* is 10^13^ and 10^17^ cm^−3^, respectively. The filled area between the red and blue lines represents the difference between the two curves. It can be found that the filled area between the red and blue lines of device 3 is the smallest, which means that the *J-V* characteristic curve of device 3 is the least affected by the change in *N_t_*. By contrast, the *J-V* characteristic curve of device 1 shows the highest sensitivity to the change in *N_t_*. All these observations explain that when *N_t_* changes in the same range, the performance of device 1 changes greatly, while the performance of device 3 is relatively stable.

The simulation results confirm that in the process of gradually increasing *N_t_*, the performance parameters of all the structures show a downward trend. However, the performance of the double ETL structure is always better than that of the single ETL structure. The device with a double ETL is less sensitive to the defect density of the perovskite active layer, and the performance is more stable than those of device 1 and device 2.

### 3.3. Effects of Defect Density at Electron Transport Layer (ETL)/Perovskite Interface

Interfacial recombination between different layers is an important factor that affects the cell performance. Interface defects capture carriers, which reduce the performance and lead to a large amount of recombination. To determine the influence of the interface defect density between the ETL and perovskite active layer on the cell performance, the interface defect density *N_it_* was set to vary from 10^10^ cm^−3^ to 10^14^ cm^−3^ in this study, and the simulation results are shown in Figure 15 and Figure 16. Figure 15 shows that the *J-V* characteristics continue to deteriorate with the increase in *N_it_*, while Figure 16 shows that *V_oc_*, *J_sc_*, *FF,* and *PCE* all decrease with the increase in *N_it_*, indicating the significant impact of *N_it_* on the performances of the PSCs. The decreases in the output parameters compared with their initial values are shown in Table 4. From that table, it is obvious that device 3 has the lowest decreases in the output parameters as *N_it_* increases to 10^14^ cm^−3^.

According to Abadi et al., the increase in *N_it_* would lead to a decrease in *V_oc_* and *J_sc_* due to an increase in the recombination rate [49]. This is because a higher *N_it_* brings more traps and recombination centers [60] and hence changes the shunt resistance [56], leading to reductions in *V_oc_* and *PCE*. This inference can also be verified by inspecting the interface recombination currents plotted in Figure 17. As shown in that figure, the interface recombination current raises as *N_it_* increases. Therefore, the interface recombination rate keeps increasing, resulting in poor cell performance. *FF* decreases as *N_it_* increases because the electric field applied to the perovskite layer decreases with increasing forward bias, and thus the collection of photogenerated charge carriers associated with the electric field becomes weaker. The interface quality, namely the junction quality, is a notable parameter for high *PCE* values in PSCs, and the improvement of the junction quality would lead to a high *PCE* [50]. Thus, to achieve a high *PCE*, there must be less charge recombination at the ETL and perovskite active layer interface.

Table 4 reveals that device 3 exhibits relatively small fluctuations in its cell performance as *N_it_* changes compared with device 1 and 2. When *N_it_* changes from 10^10^ to 10^14^ cm^−3^, *PCE* of device 1 decreases by 42.66%, while *PCE* of device 3 only decreases by 23.92%. Apparently, the performance of device 3 is more stable than that of device 1 and device 2. The reason may be that the carrier concentration of a single ETL device at the interface between the ETL and the active layer is higher, which would result in a higher recombination loss than that of the double ETL device. Figure 17 and Figure 18 confirm this hypothesis. In Figure 17, the black solid line, blue solid line, and red solid line are the lines of the interface recombination currents for electrons when *N_it_* varies between 10^10^ and 10^14^ cm^−3^, respectively. It can be seen from Figure 17 that the black solid line has the greatest slope, while the red solid line has the least slope. In other words, the interface recombination current for the electrons in device 1 undergoes the most significant change, while the interface recombination current for electrons in device 3 experiences the slightest change. Moreover, the interface recombination current of device 3 is relatively lower than the current of the other two devices. In Figure 18, the red and blue lines represent the total recombination when *N_it_* is 10^10^ and 10^14^ cm^−3^, respectively. It can be clearly seen that the total recombination of device 3 is relatively low and the change in the total recombination of device 3 is quite minimal (the direct distance between the two black dotted lines is the shortest). This suggests that the performance of a single ETL device is more affected by the interface defect density than that of the double ETL device. The increase in the interface defect density, or the increase in the number of traps at the interface, would result in more recombination, which in turn leads to performance degradation. Therefore, compared with the single ETL device, the double ETL device has a higher tolerance for interface defect density.

### 3.4. Effects of Working Temperature on the Performance

PSCs are generally used in outdoor environments at temperatures exceeding 300 K [61,62]. However, the cell performance is unstable at high temperatures due to the change in the device properties, such as carrier mobility, carrier concentration, and band gap. Therefore, the working temperature has a direct impact on the cell performance. To explore the effects of the temperature *T* on the cell performance and analyze the response sensitivities of different devices to the temperature change, the *T* was set to vary between 300 K and 500 K at an interval of 50 K in this study, and the simulation results are displayed in Figure 19. From that figure, it can be seen that with the increase in *T*, the *J-V* characteristics of the cells become worse and the increase in *T* causes *V_oc_* to decrease. Compared with *J_sc_*, the change in *T* has a more evident impact on *V_oc_*. Figure 20 shows that, with the increase in *T*, *FF* continuously decreases and the change in *FF* of device 1 reaches its maximum level.

The relationship between the carrier recombination and *T* is shown in Figure 19. It can be inferred that, with the gradual increase in *T*, the carrier recombination of the perovskite active layer continues to be strengthened. Relevant studies have confirmed that the performance of PSCs would decrease at high working temperatures. Sheikh et al. [63] revealed a monotonous increase in carrier transfer resistance and a concurrent decrease in carrier recombination resistance with increasing temperature by impedance spectroscopy analysis, indicating the high recombination of carriers at high working temperatures. Figure 21 also confirms this phenomenon. From that figure, it can be observed that the carrier concentration in the perovskite active layer increases as temperature increases. Since a higher carrier concentration means a greater recombination loss, the cell performance would deteriorate gradually in consequence. It can be concluded that, as temperature increases, the carrier transfer resistance increases but the carrier recombination resistance decreases, thus leading to an enhancement of carrier recombination.

The relationship between *V_oc_* and *T* can be described as follows [64]:(19)$$\frac{dV_{oc}}{dT}=\frac{V_{oc}-\frac{E_{g}}{q}}{T}$$

It can be determined from Equation (19) that the increase in *T* causes *V_oc_* to decrease. The relationship between *V_oc_* and the diode reverse saturation current *J*_0_ is expressed as follows [64]:(20)$$V_{oc}=\frac{kT}{q}ln\left(\frac{J_{1}}{J_{0}}+1\right)$$
where *J_l_* is the photogenerated current.

Equation (20) reveals that, as *J*_0_ gets larger, *V_oc_* will get smaller. In fact, electrons gain more energy at higher temperatures, leading to an enhancement in the carrier recombination before it is collected by the electrode [62]. The temperature *T* mainly affects the carrier recombination and *J*_0_, which in turn influences *V_oc_*. As stated by Karimi [62], *J*_0_ increases as *T* goes up and the increase in *J*_0_ in turn leads to a decrease in *V_oc_*.

In summary, the increase in temperature improves the recombination rate of the device and the increase in the carrier recombination rate is the reason for the decrease in *V_oc_*. Therefore, the influence of changing the temperature on *V_oc_* is greater than that on *J_sc_*. *V_oc_* decreases as *T* increases, leading to a decreasing trend of the *PCE*. This means that the device performance will deteriorate gradually as temperature increases. Some studies [65,66,67] have confirmed that, when *T* is excessively high, properties such as carrier mobility, carrier concentration, and band gap will all be impacted, resulting in a lower *PCE* output from the cell. Figure 22 shows that the carrier concentration in the perovskite active layer increases with increasing temperature. Since a higher carrier concentration means a greater recombination loss, the cell performance would deteriorate gradually in consequence.

In Figure 23a, the red and the blue lines represent the *J-V* characteristic curves at 300 and 500 K, respectively. The filled area between the red and blue lines represents the differences between the two curves. It can be observed that, when *T* increases from 300 to 500 K, the change in the *J-V* characteristic curves of device 1 is the largest, while that of device 3 is the smallest. Similarly, Figure 23b shows that the temperature change has the greatest impact on the carrier recombination of device 1 and the least impact on device 3 (the red and the blue lines represent the total recombination curves at 300 and 500 K, respectively). When *T* changes from 300 K to 500 K, the largest change is observed in *V_oc_* of device 1, while the least change is found in *V_oc_* of device 3. The variation of *V_oc_* is influenced by carrier recombination. Figure 21 shows that the carrier recombination change of device 1 is more obvious than that of device 3. As a result, the change of *V_oc_* in device 1 is more significant. Figure 20 shows that the *FF* of the device decreases as the temperature increases. In comparison, the *FF* change of device 1 is the largest, while that in device 3 is the smallest. *FF* measures the series resistance of the device. Therefore, it can be inferred that the series resistance change in device 1 is the largest, while that in device 3 is the least. Photovoltaic characteristics can be represented by *J-V* characteristic curves, which contain key information such as *V_oc_*, *J_sc_*, *FF*, and *PCE*. Thus, the most significant changes in *V_oc_* and *FF* of device 1 result in the most notable variations in its *J-V* characteristics curve. It can be calculated from Figure 23c that the *PCE* values of device 1, device 2, and device 3 decrease by 94%, 61%, and 52%, respectively, as *T* is gradually increased from 300 K to 500 K. Device 3 has the least performance degradation. All these results indicate that device 1 is most sensitive to the temperature change, while device 3 maintains the highest stability under the changing temperature.

## 4. Conclusions

In this study, a double ETL structure of FTO/TiO_2_/ZnO/(FAPbI_3_)_0_._85_(MAPbBr_3_)_0.1_/Spiro-OMeTAD is investigated by comparing its performance with those of single ETL structures of FTO/TiO_2_/(FAPbI_3_)_0.85_(MAPbBr_3_)_0.1_/Spiro-OMeTAD and FTO/ZnO/(FAPbI_3_)_0.85_(MAPbBr_3_)_0.1_/Spiro-OMeTAD using SCAPS-1D software. The defect density of the perovskite active layer and the defect density at the interface between the ETL and the perovskite active layer, and the different temperatures were studied and compared, while keeping the remaining parameters fixed. The simulation results reveal that the double ETL structure enables a simultaneous control of the conduction band offset between the perovskite active layer and the ETL as well as the energy level difference between the ETL and the electrode, forming a stepped conduction band structure and reducing the non-radiative recombination caused by the energy level dislocation. The increase in the defect density in the perovskite active layer would lead to the formation of more carrier recombination centers, which facilitates the carrier recombination. Benefiting from the reduced number of traps at the interface due to the low defect density in the interface between the ETL and the perovskite active layer, interface recombination is low, resulting in a superior performance. The increase in the temperature would improve the recombination rate of the device, thus causing *V_oc_* and *PCE* to decrease. In terms of the defect density of the perovskite active layer, the defect density at the interface between the ETL and perovskite active layer, and the cell performance under different working temperatures, the performance of the double ETL PSC is superior to that of the single ETL PSC and its stability is also higher. The findings of this research clearly affirm the possibility of preparing a stable perovskite solar cell.

## Figures and Tables

**Figure 1 nanomaterials-13-01313-f001:**
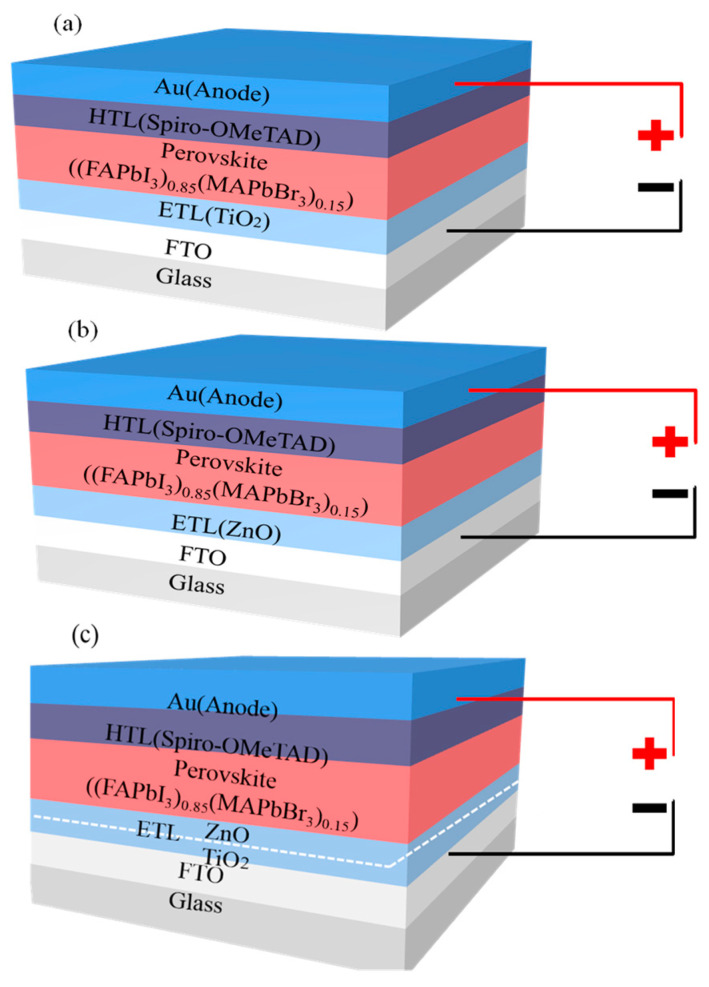
Solar cell structures of different electron transport layers (ETLs): (**a**) solar cell structure of TiO_2_ as ETL, (**b**) solar cell structure of ZnO as ETL and (**c**) solar cell structure of TiO_2_/ZnO as ETL.

**Figure 2 nanomaterials-13-01313-f002:**
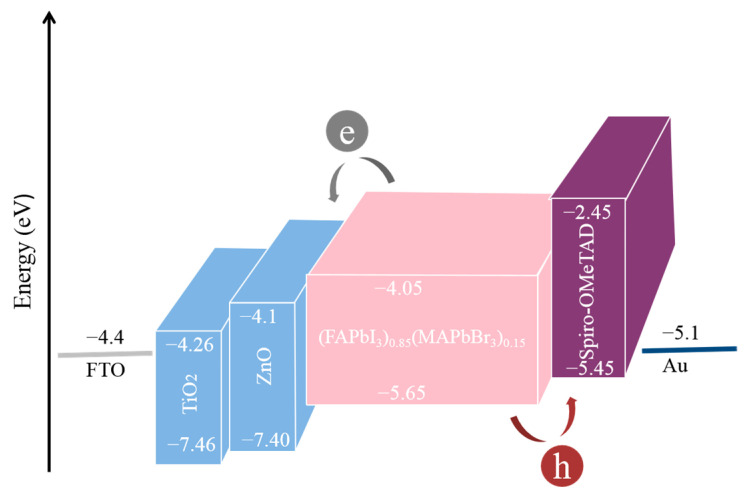
Energy level diagram.

**Figure 3 nanomaterials-13-01313-f003:**
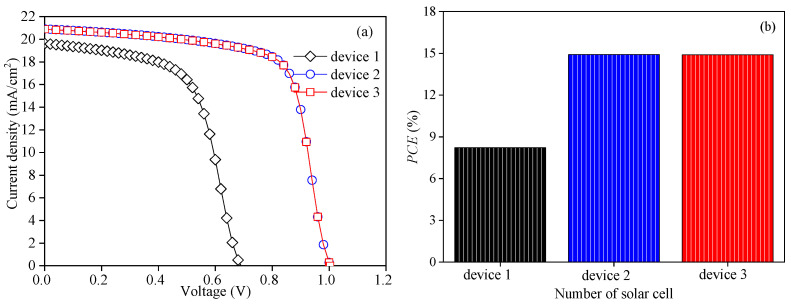
Performance comparison of three initial models: (**a**) *J-V* characteristic curves of three initial models, (**b**) *PCE* of three initial models, (**c**) quantum efficiency *(QE)* of three initial models, and (**d**) conduction band of device. (the area circled by the green dotted line represents the interface energy band shift between the ETL and perovskite active layer, and the area circled by the orange dotted line represents the barrier formed between the ETL and the cathode).

**Figure 4 nanomaterials-13-01313-f004:**
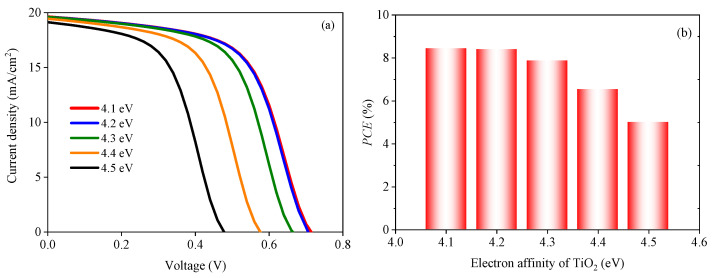
Effect of the electron affinity of TiO_2_ on the solar cell: (**a**) *J-V* characteristic curves under different electron affinities of TiO_2_ and (**b**) power conversion efficiency *(PCE)* under different electron affinities of TiO_2_.

**Figure 5 nanomaterials-13-01313-f005:**
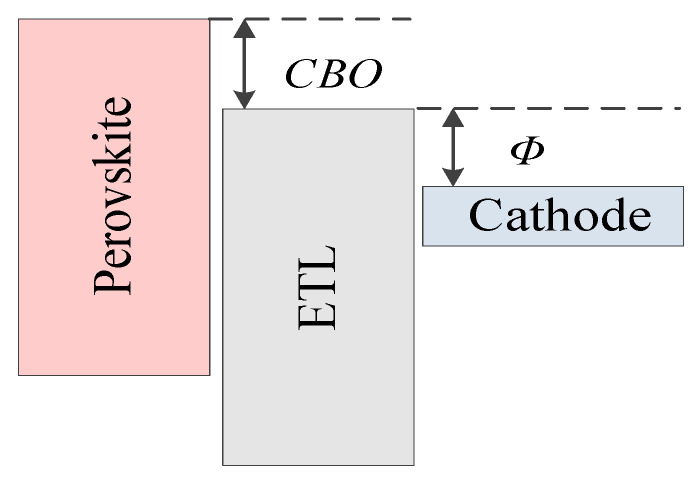
Schematic diagram of conduction band offset (*CBO*) and *Φ*.

**Figure 6 nanomaterials-13-01313-f006:**
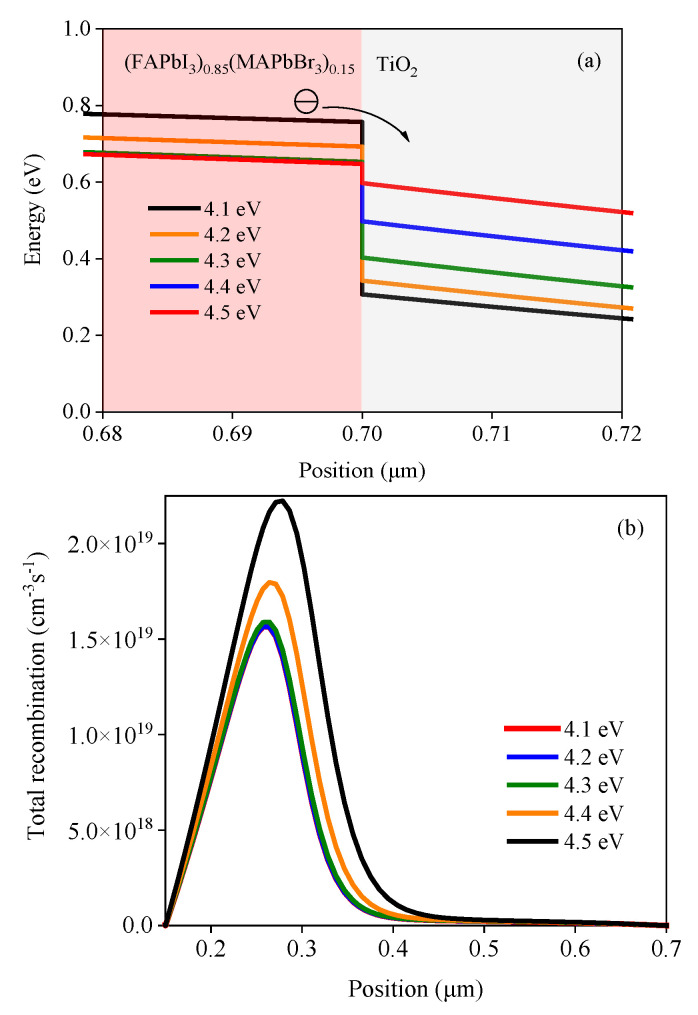
Effect of the electron affinity of TiO_2_ on the band structure and total recombination: (**a**) band structure under different electron affinities of TiO_2_ and (**b**) total recombination under different electron affinities of TiO_2_.

**Figure 7 nanomaterials-13-01313-f007:**
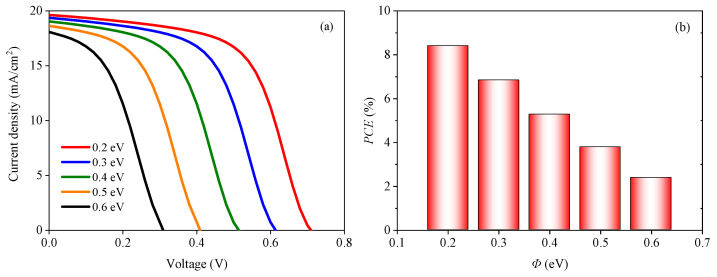
Effect of *Φ* on the solar cell: (**a**) *J-V* characteristic curves with different *Φ* values and (**b**) *PCE* with different *Φ* values.

**Figure 8 nanomaterials-13-01313-f008:**
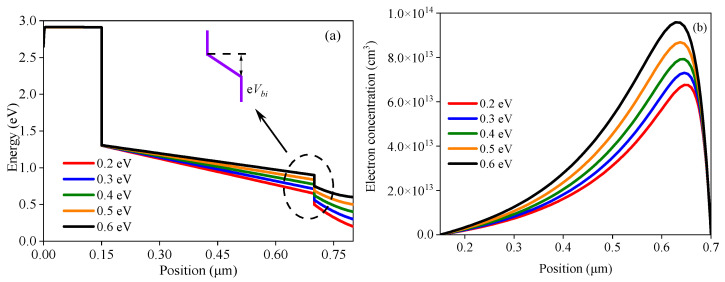
Effect of *Φ* on the band structure and total recombination: (**a**) band structures under different *Φ* values, (**b**) total recombination with different *Φ* values.

**Figure 9 nanomaterials-13-01313-f009:**
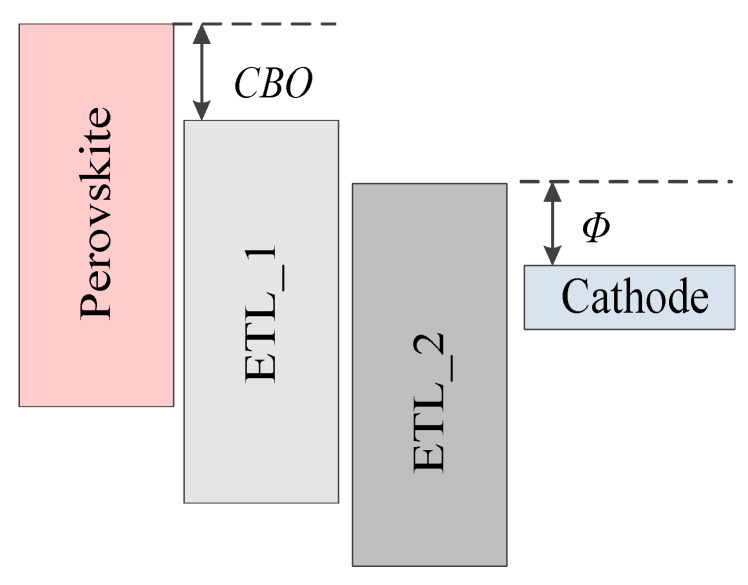
Schematic diagram of conduction band offset (*CBO*) and *Φ* in a double ETL structure.

**Figure 10 nanomaterials-13-01313-f010:**
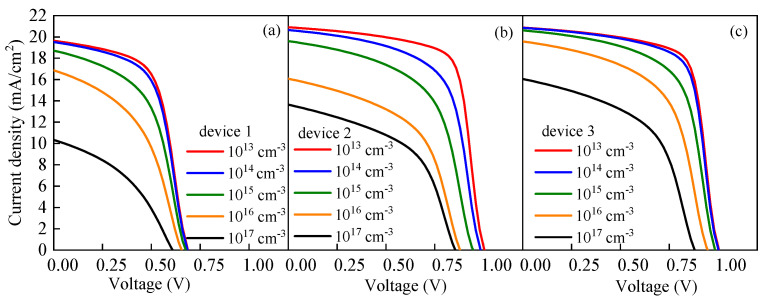
Effect of Nt on the *J–V* characteristic curve: (**a**) effect of *N_t_* on *J-V* characteristic curve of device 1, (**b**) effect of *N_t_* on *J-V* characteristic curve of device 2, and (**c**) effect of *N_t_* on *J-V* characteristic curve of device 3.

**Figure 11 nanomaterials-13-01313-f011:**
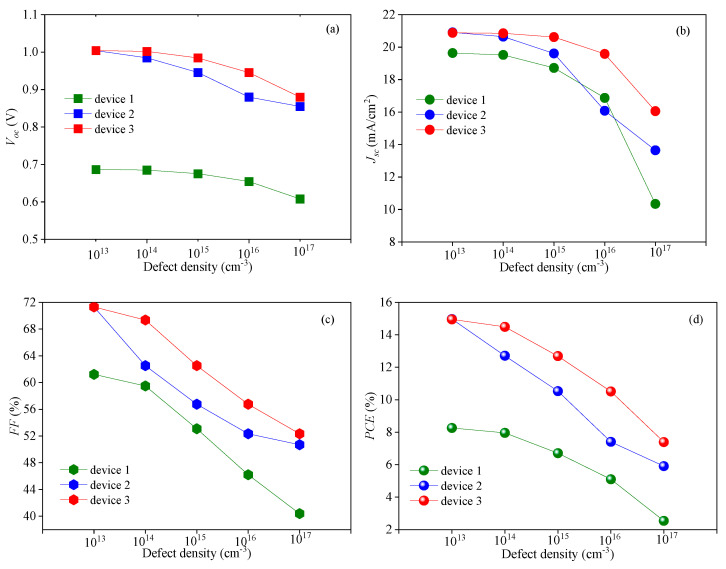
Effect of *N_t_* on the cell output parameters: (**a**) effect of *N_t_* on *V_oc_*, (**b**) effect of *N_t_* on *J_sc_*, (**c**) effect of *N_t_* on *FF*, and (**d**) effect of *N_t_* on *PCE*.

**Figure 12 nanomaterials-13-01313-f012:**
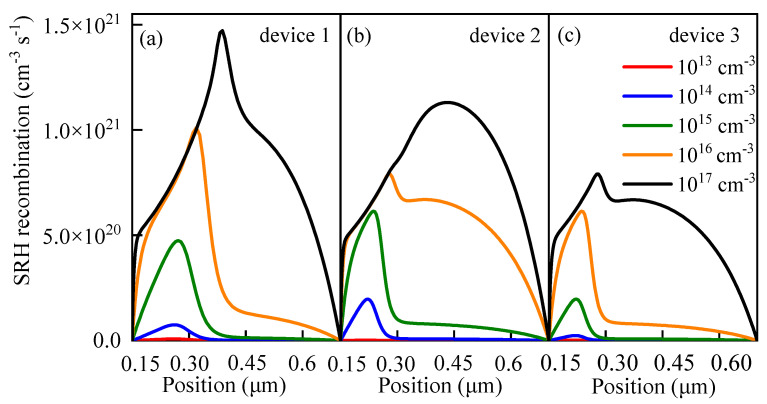
Relationship between *N_t_* and *R^SRH^*: (**a**) relationship between *N_t_* and *R^SRH^* in device 1, (**b**) relationship between *N_t_* and *R^SRH^* in device 2, and (**c**) relationship between *N_t_* and *R^SRH^* in device 3.

**Figure 13 nanomaterials-13-01313-f013:**
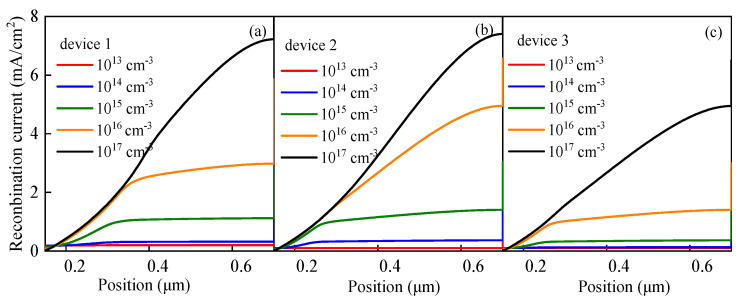
Recombination current at different *N_t_*: (**a**) recombination current at different *N_t_* in device 1, (**b**) recombination current at different *N_t_* in device 2, and (**c**) recombination current at different *N_t_* in device 3.

**Figure 14 nanomaterials-13-01313-f014:**
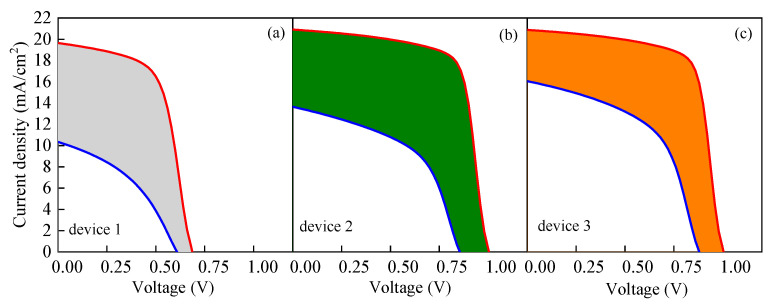
Change of *J-V* characteristic curve when *N_t_* is 10^13^ and 10^17^ cm^−3^: (**a**) change in *J-V* characteristic curve of device 1, (**b**) change in *J-V* characteristic curve of device 2, and (**c**) change in *J-V* characteristic curve of device 3.

**Figure 15 nanomaterials-13-01313-f015:**
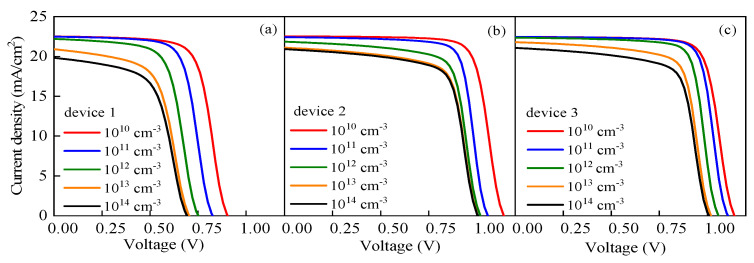
Effects of *N_i__t_* on the cell performance: (**a**) effect of *N_i__t_* on *J-V* characteristic curve of device 1, (**b**) effect of *N_i__t_* on *J-V* characteristic curve of device 2, and (**c**) effect of *N_i__t_* on *J-V* characteristic curve of device 3.

**Figure 16 nanomaterials-13-01313-f016:**
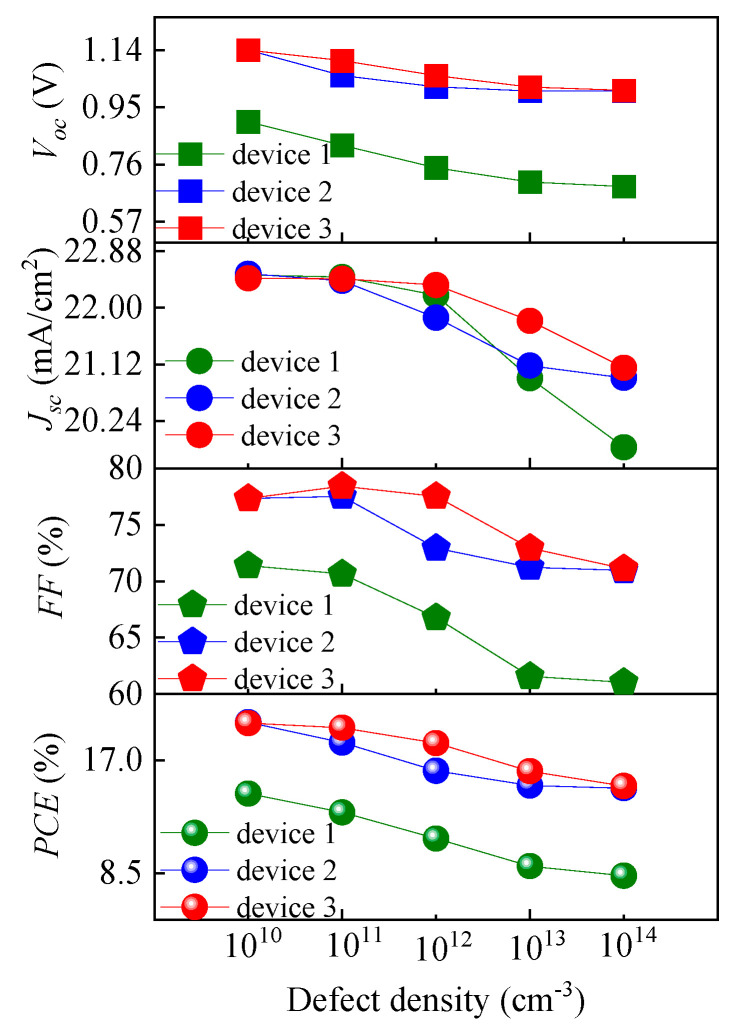
Effects of *N_i__t_* on the cell output parameters.

**Figure 17 nanomaterials-13-01313-f017:**
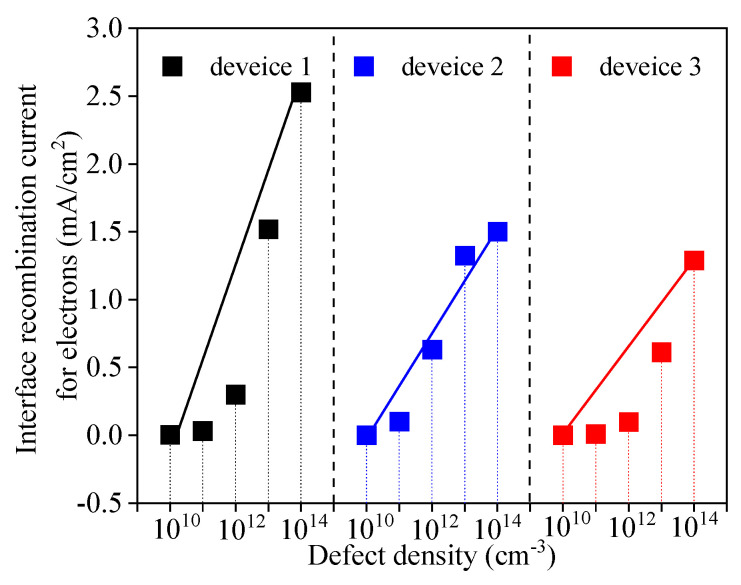
Effects of *N_i__t_* on the interface recombination current for electrons.

**Figure 18 nanomaterials-13-01313-f018:**
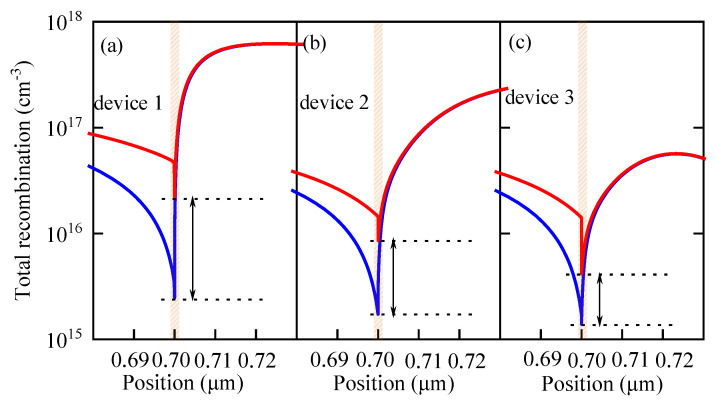
Effects of *N_i__t_* on the interface recombination: (**a**) effect of *N_it_* on the interface recombination of device 1, (**b**) effect of *N_it_* on the interface recombination of device 2 and (**c**) effect of *N_it_* on the interface recombination of device 3.

**Figure 19 nanomaterials-13-01313-f019:**
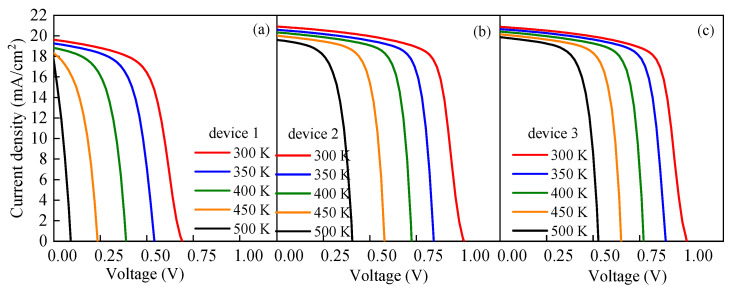
Effects of the temperature on the cell performance: (**a**) *J-V* characteristics of device 1 at different temperatures, (**b**) *J-V* characteristics of device 2 at different temperatures, and (**c**) *J-V* characteristics of device 3 at different temperatures.

**Figure 20 nanomaterials-13-01313-f020:**
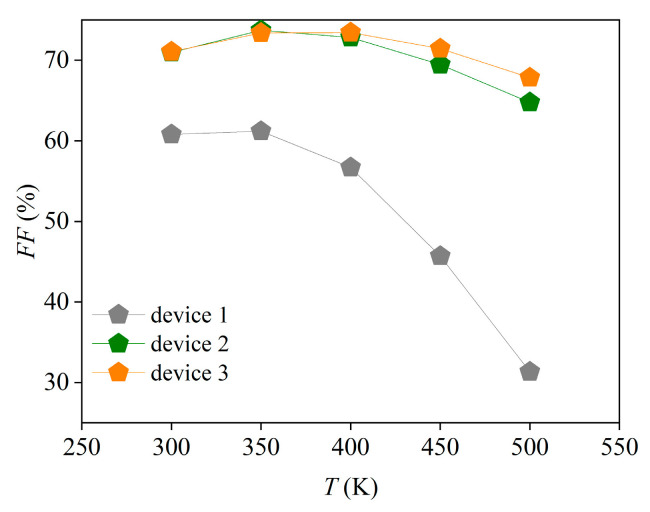
Effects of temperature on the cell fill factor.

**Figure 21 nanomaterials-13-01313-f021:**
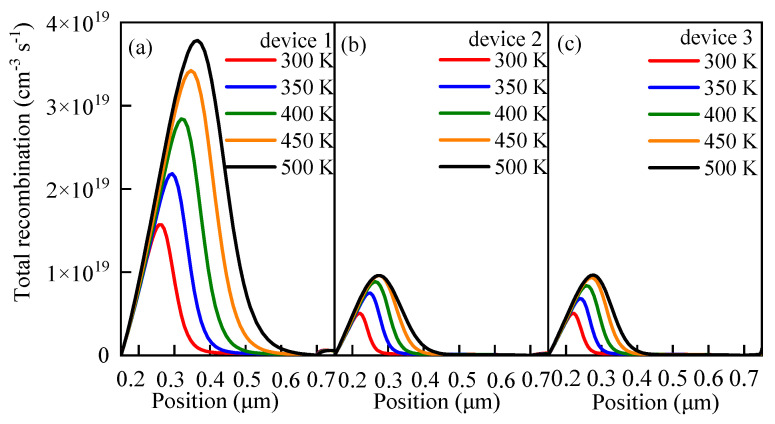
Effects of temperature on the carrier recombination: (**a**) carrier recombination of device 1 at different temperatures, (**b**) carrier recombination of device 2 at different temperatures, and (**c**) carrier recombination of device 3 at different temperatures.

**Figure 22 nanomaterials-13-01313-f022:**
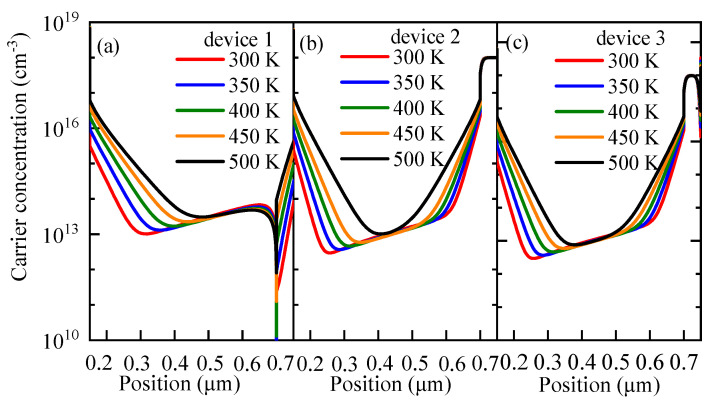
Effects of temperature on the carrier concentration: (**a**) carrier concentration of device 1 at different temperatures, (**b**) carrier concentration of device 2 at different temperatures, (**c**) carrier concentration of device 3 at different temperatures.

**Figure 23 nanomaterials-13-01313-f023:**
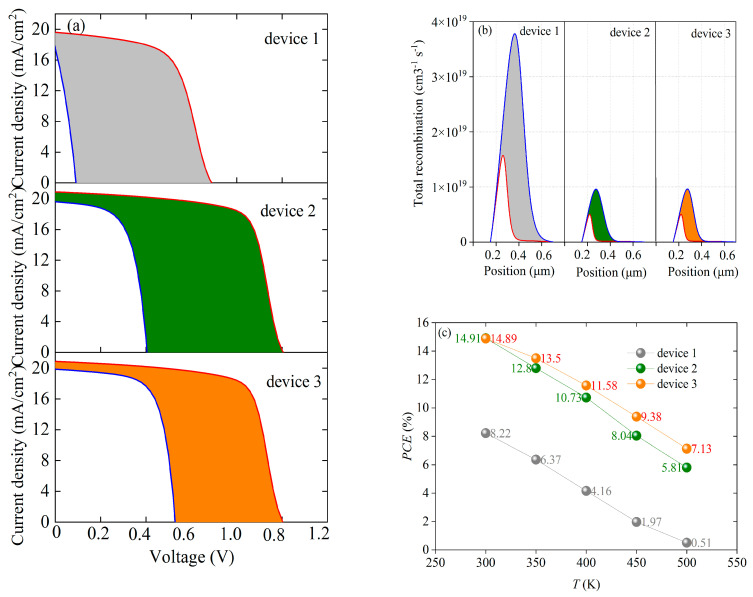
Performance fluctuations of three devices: (**a**) changes in *J-V* characteristic curves, (**b**) changes in total recombination, (**c**) trends of *PCE* values with temperature.

**Table 1 nanomaterials-13-01313-t001:** Basic physical parameters of the various layers.

Parameters	TiO_2_ [44,45,46]	ZnO [47,48]	(FAPbI_3_)_0.85_(MAPbBr_3_)_0.15_ [49]	Spiro-OMeTAD [27,50]
Thickness (nm)	--	--	550	150
Bandgap *E_g_* (eV)	3.2	3.3	1.6	3.0
Electron affinity χ (eV)	4.26	4.1	4.05	2.45
Dielectric permittivity *ε_r_*	9.0	9.0	30	3.0
CB effective density of states *N_c_* (cm^−3^)	2 × 10^18^	4 × 10^18^	2.2 × 10^18^	2.2 × 10^18^
VB effective density of states *N_v_* (cm^−3^)	1.8 × 10^19^	1 × 10^19^	1.8 × 10^19^	1.8 × 10^19^
Electron mobility *μ_n_* (cm^2^/Vs)	20	100	0.1261	2 × 10^−4^
Hole mobility *μ_p_* (cm^2^/Vs)	10	25	126.9	2 × 10^−4^
Donor concentration *N_D_* (cm^−3^)	1 × 10^16^	1 × 10^18^	1 × 10^13^	0
Acceptor concentration *N_A_* (cm^−3^)	0	0	1 × 10^13^	1 × 10^19^
Defect density *N_t_* (cm^−3^)	1 × 10^15^	1 × 10^15^	2.16 × 10^13^	1 × 10^15^

**Table 2 nanomaterials-13-01313-t002:** Interface characteristics setting.

Interface Characteristics	TiO_2_/PSC	PSC/Spiro-OMeTAD
Defect type	Neutral	Neutral
Capture cross section electrons (cm^2^)	2 × 10^−14^	2 × 10^−14^
Capture cross section holes (cm^2^)	2 × 10^−14^	2 × 10^−14^
Energetic distribution	Gaussian	Gaussian
Reference for defect energy level *E_t_*	Above the highest *E_v_*	Above the highest *E_v_*
Energy with respect to Reference (eV)	0.6	0.6
Characteristic energy (eV)	0.1	0.1
Total density (cm^−3^)	2 × 10^19^	2 × 10^19^

**Table 3 nanomaterials-13-01313-t003:** Decreases of output parameters.

Parameters	*V_oc_*	*J_sc_*	*FF*	*PCE*
Device 1	12.37%	47.33%	34.02%	69.25%
Device 2	14.85%	34.74%	28.89%	60.52%
Device 3	11.47%	23.10%	26.60%	50.57%

**Table 4 nanomaterials-13-01313-t004:** Decreases in output parameters.

Structure	*V_oc_*	*J_sc_*	*FF*	*PCE*
Device 1	23.83%	11.90%	14.48%	42.66%
Device 2	11.90%	7.18%	8.25%	25.00%
Device 3	11.75%	6.19%	8.04%	23.92%

## Data Availability

The data that support the findings of this study are available from the corresponding author upon reasonable request.

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
