# Peer review of "Numerical Analysis of Stable (FAPbI3)0.85(MAPbBr3)0.15-Based Perovskite Solar Cell with TiO2/ZnO Double Electron Layer"

_nanomaterials, 2023, doi:10.3390/nano13081313_

Round 1

Reviewer 1 Report

This work shows that a double ETL structure of FTO/TiO2/ZnO/(FAPbI3)0.85(MAP- 474 bBr3)0.1/Spiro-OMeTAD is investigated by comparing its performance with those of single ETL structures of FTO/TiO2/(FAPbI3)0.85(MAPbBr3)0.1/Spiro-OMeTAD and FTO/ZnO/(FAPbI3)0.85(MAPbBr3)0.1/Spiro-OMeTAD using SCAPS-1D software. The defect  density of the perovskite active layer, the defect density at the interface between the ETL and the perovskite active layer, and temperature were studied and compared. Herein, the reviewer gives some comments for the authors' consideration, and the authors need to explain my questions before its accept.

Comments:

1. The simulation results reveal that the double ETL structure enables a simultaneous control of the conduction band offset between the perovskite active layer and the ETL as well as the energy level difference between the ETL  and the electrode, and the detailed mechanism for this should be given.

2. It can be observed that when T increases from 300 to 450 500 K, the change in the J-V characteristic curves of device 1 is the largest, while that in device 3 is the smallest, why?

3. PSCs have been widely reported, and recent works about high performance PSCs should be added, such as, 10.1016/j.esci.2021.09.005 ; 10.1007/s40820-022-00992-5; 10.1016/j.esci.2022.04.001.

4. It 416 can be inferred that with the gradual increase in T, the carrier recombination of the per- 417 ovskite active layer continues to be strengthened, how to understand this?

Author Response

We want to express our deep gratitude to the reviewer for taking time to evaluate this paper and putting forth expert advice and insightful comments. We have tried our best to modify this paper to address all those comments. Please review the revised manuscript and let us know if any further revisions are needed. Thank you very much!

Reviewer 2 Report

The article used SCAPS-1D to compare three electron transporting layers (ETL) TiO2, ZnO and TiO2/ZnO for perovskite solar cells to reveal that the double ETL TiO2/ZnO reduces the energy level dislocation to inhibit the non-radiative recombination, and is also more tolerant to defect density and temperature. These main points are supported by the evidence in this study, given the parameters used to simulate the solar cells. The TiO2/ZnO double ETL layer explored in this study provided a valuable datapoint in improving the perovskite solar cell performance, as contact engineering is crucial to achieve high performance and stability of solar cells. 

However, I have concerns about the given parameters in this study used to simulate the solar cells. In the simulation results, TiO2 based solar cells (device 1) only has <10% PCE according to Figure 10. This is much lower than what's widely reported in literature for lead based perovskite solar cells. The parameters used for simulation is from citation 17 and 31-36 according to this manuscript, but in all of those citations the lead based perovskite solar cells have much higher PCE than 10%. Furthermore, a very similar study with similar premise from 2022 (10.21272/jnep.14(1).01033) reported a much higher performance with TiO2. Since the baseline scenario of TiO2 based cell performance is much lower than expected, the manuscript needs an in-depth discussion to explain why this is happening, point out the simulation parameter leading to this unexpected low PCE, and justify that choice. Without a proper discussion on this issue, I do not consider the baseline TiO2 scenario to be properly configured, and any conclusion from comparing to the baseline scenario should not be trusted.

In page 6 Table 1, the thickness of TiO2 and ZnO layers are omitted. Is there any reason why the thicknesses are not reported? In page 6 line 187 the manuscript claims the absorption of photons by device 1 is inadequate. Why are the absorption of photons different across different devices, shouldn't the three devices use the same perovskite absorber layer? 

Due to the aforementioned weakness, I recommend reconsidering after major revisions.

Author Response

(The authors gave the same response as above.)

Reviewer 3 Report

I am happy to recommend this paper for publication.

Author Response

Dear Reviewer,

Thank you so much for your recommendation, we really appreciate your time in reviewing this paper and your favorable consideration.

Sincerely Yours,

Yucheng

Reviewer 4 Report

This manuscript proposes the numerical analysis of stable (FAPbI3)0.85(MAPbBr3)0.15-based perovskite solar cell with TiO2/ZnO double electron layer. The topic is interesting, and certainly consistent with the contents to be proposed to the readers of “Nanomaterials”. Moreover, the manuscript is well written and can be read with pleasure: this represents an important aspect in the current scenario of publications in international journals. Overall, I think that this manuscript has to be accepted, but the Authors should take into account the following minor revisions (in terms of bibliographic updates, grammar corrections and content deepening):

-          Detailed revisions: I spent several hours reading this manuscript, and Authors are asked to follow carefully the attached PDF file where I highlighted some points to be addressed. The attached file also contains language mistakes and typos; some questions related to manuscript contents could also be present and Authors must consider them properly before submitting the revised manuscript. A point-by-point reply is required when the revised files are submitted.

-          The Introduction should give a wider overview on the present scenario related to new trends in PSCs, both in terms of recently published reviews and research articles. In particular, sustainable materials and integrated devices are missing and a paragraph on this topic is highly suggested to be added in the Introduction. Authors are invited to go through the literature published in the last six months on these issues, and also on concepts developed some years ago in this field. Some of them are also mentioned in the above mentioned PDF file.

-          Authors should provide a clear explanation on the experimental error of the proposed research work. In particular, reproducibility of the phenomena described in the manuscript should be clearly stated in the “Results and Discussion” section; besides, some notes in the “Materials and Methods” section should be added highlighting which kind of experimental approach has been followed to check the reproducibility of the proposed system, the latter being of noteworthy importance in the present research field.

Author Response

We want to express our deep gratitude to the reviewer for taking enormous time to go through almost every detail of this paper and putting forth expert advice and insightful comments. We have tried our best to modify this paper to address all those comments. Please review the revised manuscript and let us know if any further revisions are needed. Thank you very much!

Round 2

Reviewer 2 Report

The authors have addressed most of my concerns in this version. One concern remaining is the low solar cell performance in the reference cell with TiO2 as the electron transporting layer. The low performance of < 10%  is significantly below what's reported in literature. The question is whether this beneficial effect of using TiO2/ZnO double layer observed in this manuscript will still hold when applied to perovskite layers with much better quality that lots of other groups are producing. 

Author Response

Thank you so much for your insightful comments. Some authoritative reports have confirmed that double ETL structure can improve solar cell performance. Qiu et al. [S1] used the PCBM/ZnO double ETL to construct CH3NH3PbIxCl3-x based perovskite solar cells. They proposed that double ETLs was helpful to significantly improve the device performance, which has not been mentioned in previous reports. TiO2/SnOxCly double layer was employed as the ETL by Wu et al. [S2] for planar perovskite solar cells. Compared with bare TiO2, perovskite solar cell based on TiO2/SnOxCly showed drastically improved PCE and reduced hysteresis. et al. To achieve high Voc and PCE, Wang et al. [S3] introduced ZnO/SnO2 double electron transport layer for CH3NH3PbI3-based PSCs, Mohamadkhani et al. [S4] reported that PCE of planar PSCs fabricated using SnO2/CdS as ETL was increased compared with that using SnO2 as ETL. [S5] demonstrated a high-performance planar heterojunction PSCs with PCBM/N2200 as double electron transport layers. They claimed that the PCBM/N2200 double ETLs increase built-in potential of devices and decrease interfacial energy barrier of MAPbI3/PCBM and then result in higher Voc. Gao et al. [S6] believed that interface charge extraction was improved by double electron transport layers for high-efficient planar PSC. Khan et al. [S7] introduced a double electron transport layer that consists of SnO2/ZnO and use it to prepare MAPbI3-based planar heterojunction PSCs for mitigating the energy loss.  Although there are not many reports about perovskite solar cells with double electron transport layers, the literature mentioned above all confirmed that the double electron transport layers can promote the solar cell performance. Therefore, perovskite solar cells with double electron transport layers have a higher tolerance to defect density and working temperature compared with that with single electron transport layer proposed by this paper is credible.

A proper ETL is helpful to conduct electrons and block holes and reduce energy loss. Thus, it should have a high conductivity and an excellent band alignment with the perovskite active layer. Therefore, adopting TiO2/ZnO double ETLs also can promote cell performance if the band alignment between TiO2/ZnO double ETLs  and the perovskite active layer is excellent.

[S1]Qiu, W.; Buffiere, M.; Brammertz, G.; Paetzold, U. W.; Froyen, L.; Heremans, P.; Cheyns, D.  High efficiency perovskite solar cells using a PCBM/ZnO double electron transport layer and a short air-aging step. Org. Electron, 2015.26, 30-35. https://doi.org/10.1016/j.orgel.2015.06.046.

[S2]Wu, C.; Huang, Z.; He, Y.; Luo, W.; Ting, H.; Li, T.; Xiao, L.  TiO2/SnOxCly double layer for highly efficient planar perovskite solar cells. Org. Electron, 2017, 50, 485-490.  https://doi.org/10.1016/j.orgel.2017.07.050.

[S3]Wang, D.; Wu, C.; Luo, W.; Guo, X.; Qu, B.; Xiao, L.; Chen, Z.. ZnO/SnO2 double electron transport layer guides improved open circuit voltage for highly efficient CH3NH3PbI3-based planar perovskite solar cells. ACS Applied Energy Materials, 2018, 1, 2215-2221. https://doi.org/10.1021/acsaem.8b00293.

[S4]Mohamadkhani, F.; Javadpour, S.; Taghavinia, N. Improvement of planar perovskite solar cells by using solution processed SnO2/CdS as electron transport layer. Sol. Energy, 2019, 191, 647-653.https://doi.org/10.1016/j.solener.2019.08.067.

[S5]Ren, C.; He, Y.; Li, S.; Sun, Q.; Liu, Y.; Wu, Y.; Wu, Y. Double electron transport layers for efficient and stable organic-inorganic hybrid perovskite solar cells. Org. Electrons, 2019, 70, 292-299.

[S6]Gao, Y.; Wu, Y.; Liu, Y.; Chen, C.; Shen, X.; Bai, X.; Zhang, Y.  Improved Interface Charge Extraction by Double Electron Transport Layers for High-Efficient Planar Perovskite Solar Cells. Solar RRL, 2019, 3, https://doi.org/1900314.10.1002/solr.201900314.

[S7]Khan, U.; Iqbal, T.; Khan, M.; Wu, R. SnO2/ZnO as double electron transport layer for halide perovskite solar cells. Sol Energy, 2021, 223, 346-350. https://doi.org/10.1016/j.solener.2021.05.059.